# ENTROPY VOTING BETWEEN CAPSULES

## ABSTRACT

Capsule networks offer a promising solution in computer vision by addressing the limitations of convolutional neural networks (CNNs), such as data dependency and viewpoint challenges. Unlike CNNs, capsules reduce the need for data augmentation by enhancing generalization from limited training data. We explore capsules from the perspective of information theory, viewing them as Monte Carlo sampled continuous random variables. We use marginal differential entropy to measure the information content of capsules, and relative entropy to model the agreement between lower-level and higher-level capsules. The proposed entropy voting method aims to maximize capsule marginal entropies and to minimize their relative entropy. We show that our approach performs better or comparably against state-of-the-art capsule networks while significantly improving inference time. This research highlights the synergy between capsules and information theory, providing insights into their combined potential.

## 1 INTRODUCTION

Capsule networks have emerged as a promising approach in computer vision, with a clear focus on alleviating the shortcomings of convolutional neural networks (CNNs). CNNs frequently struggle with several issues, notably the need for enormous amounts of data, the vulnerability to challenges posed by varying viewpoints, scaling, occlusion, and deformation in images, and the lack of understanding of spatial relationships. To circumvent these limitations, CNNs require an extensive use of data augmentation techniques. In contrast, capsules are designed to reduce the reliance on data augmentation, aiming to directly mitigate the need for such techniques and enhance the network's ability to generalize from limited training data.

Capsules, first proposed by Hinton et al. (2011) and later refined by Sabour et al. (2017), can be understood as vector-valued functions, designed to capture rich and hierarchical information about objects in an image. Unlike traditional convolutional network units - which are scalar-valued functions - capsules explicitly encode object instantiation parameters, such as texture and pose, thus enabling them to learn more robust representations without having to use data augmentation to achieve the same effect. The intuition behind capsule networks is to model objects as a composition of capsules, with lower-level capsules encoding parts of an object and higher-level capsules encoding the object as a whole, which are then linked together through a voting procedure that is commonly referred to as agreement between lower- and higher-level capsules.

The different levels of capsules provide a modular design that, through agreement, is able to learn part-whole relationships and spatial hierarchy in the image. Thus, capsules enable the network to utilize both global and local visual information at various levels of abstraction, thereby encoding a more comprehensive representation of the input image. While state-of-the-art methods mainly focus on modelling agreement between the capsules, we investigate capsule networks from the point of view of information theory, exploring how information-theoretic concepts can enhance our understanding and utilization of capsules.

Viewed through the lens of information theory, capsules can be considered as Monte Carlo sampled continuous random variables. Thus, marginal differential entropy serves as a crucial indicator of the quality of information encoded by each capsule. When these marginal entropies are maximized, capsules become adept at capturing more comprehensive and descriptive information. Furthermore, the agreement between lower- and higher-level capsules can be effectively modeled using relative entropy, also known as Kullback-Leibler divergence. Our proposed approach, called entropy voting,

is designed with the dual objective of maximizing the marginal entropies of capsules and minimizing the relative entropy between lower- and higher-level capsules.

In our experiments, we provide empirical evidence that underscores the effectiveness of these information-theoretic principles in elevating the performance and robustness of capsule networks. This investigation illuminates the synergistic relationship between capsule networks and information theory, shedding valuable insights into their combined potential. We compare our results against three baselines, which are the state-of-the-art of capsule networks.

## 2 RELATED WORKS

The concept of capsules was first introduced by Hinton et al. (2011) as a way to encode both descriptor and probability parameters of an object or object part. However, it was the work by Sabour et al. (2017) that formulated the concept of capsule networks as it is now known, including (1) the dynamic routing mechanism, (2) a capsule-specific loss function, (3) a reconstruction regularizer. and (4) a squash activation function that converts the magnitude of each parent capsule to a probability. Since then, there have been many works on capsule networks, namely focusing on improving the underlying routing algorithm, but also some modeling capsules as matrices rather than vectors.

Hinton et al. (2018) proposed to treat capsules as $4 \times 4$ pose matrices with an activation probability. The method uses an iterative expectation-maximization algorithm that includes a transformation matrix which learns a mapping between capsules at different levels. In addition, the authors propose a new, matrix-capsule specific loss function.

Choi et al. (2019) investigate a routing mechanism based on attention and propose a modification to the squash activation function. Routing based on self-attention has also been proposed by Mazzia et al. (2021). They, too, introduced a modification to the squash activation function, similar to Choi et al. (2019).

Routing in capsule networks has also been investigated as an optimization problem by Wang and Liu (2018). The authors formulated routing as a clustering problem, and used a Kullback-Leibler divergence term for regularization. In their proposed work, capsules are squashed only after routing to stabilize the growth of capsule under the assumption that only parent capsules should have a probabilistic magnitude.

Zhang et al. (2018) investigate an alternative approach to capsule networks, based on orthogonal representation of capsule subspaces onto which feature vectors (i.e. first-level capsules) are projected. Each capsule subspace is updated until it contains input feature vectors corresponding to the associated class, i.e. parent capsule. Their proposed method does not involve any routing mechanism as capsule subspaces and projection matrices are learned through back-propagation.

Ribeiro et al. (2020) propose a routing method based on variational Bayesian inference, an interpretation of expectation-maximization. In their proposed method capsule activation is done only after routing. Another alternative routing method is proposed by Zhang et al. (2020), which is based on kernel density estimation. The authors proposed two different approaches, one based on expectation-maximization, and the other on mean shift, a feature-space analysis technique. Building on variational methods, De Sousa Ribeiro et al. (2020) investigate a routing mechanism based on variational inference that accounts for uncertainty.

Hahn et al. (2019) investigate routing as a separate network. Each capsule is routed independently by a dedicated sub-network, each of which is a single-layer perceptron (SLP), meaning that there are as many SLPs as there are higher-level capsules. The SLPs have two sets of weights, one for pose and one for routing. Rajasegaran et al. (2019) explore a routing method based on 3D convolutions, and introduce a 3D variant of the squash function.

However, Byerly et al. (2021) show that no routing is needed by using homogeneous vector capsules, which replace matrix multiplication with the Hadamard product, in order to keep capsule dimensions untangled. Their approach also leverages multiple classification branches.

The aforementioned approaches fall short in at least one of two aspects; computationally inefficient routing (be it iterative, attention-based, a sub-network, or other), and capsule activation done using some non-differentiable function (the squash function or a variant thereof, such as vector norm). The

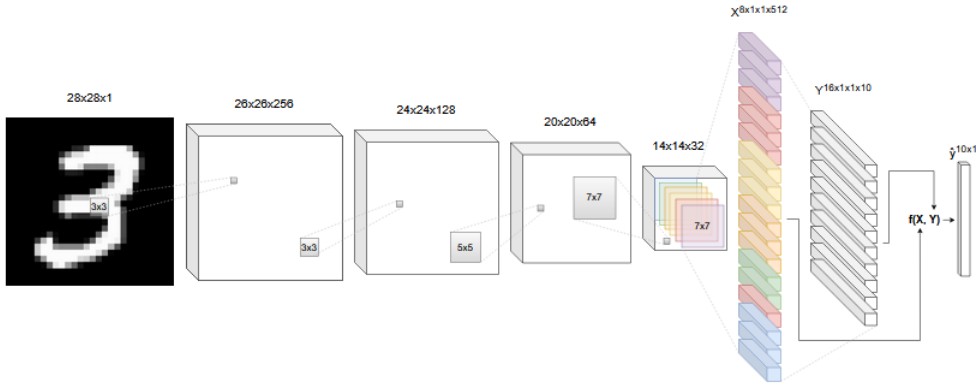

Figure 1: An abstract illustration of the network architecture w.r.t. MNIST. f(X, Y) denotes the entropy voting function.

first aspect leads to large execution times during training and/or inference, and the second aspect often leads to instability during training. Moreover, in all cases test performance is reported using significant amounts of data augmentation. Keeping in mind that one of the key ideas of capsule networks is to reduce the necessity of data augmentation, this is at least somewhat questionable. An objective evaluation of the performance of capsule networks w.r.t. the issues they are designed to mitigate should not be based on data augmentation. We address all three aspects in our work. Furthermore, we assume that the large execution times (not only for training but especially for inference) are the main reason that capsule networks are still rarely used, despite their theoretical advantages, which is not addressed by any of the aforementioned works.

We selected Sabour et al. (2017), Mazzia et al. (2021), and Byerly et al. (2021) as our baselines due to the fact that they are the state-of-the-art capsule networks on MNIST, which is still the de facto dataset for capsule networks.

## 3 METHODOLOGY

### 3.1 NETWORK ARCHITECTURE

The overall architecture of our method is illustrated in Figure 1. The first four layers of the network are simple 2D convolutional layers with batch normalization and the ReLU activation function. In the first two layers, kernel size of $3 \times 3$ is used, followed by kernel of size $5 \times 5$ in the subsequent layer to extract local features such as corners and edges, and other fine-grained details. In the final convolutional layer, a $7 \times 7$ kernel is used to capture global, more abstract features such as object shape and texture. Note that the architecture as illustrated in Figure 1 is w.r.t. MNIST, and thus the tensor dimensions are subject to change depending on the dataset.

Using smaller kernels followed by larger kernels while reducing the overall number of kernels provides computational benefits. Smaller kernels have fewer parameters and by gradually increasing kernel size, one can reduce the spatial resolution of the feature maps while maintaining important global information. In addition, decreasing the number of kernels linearly forces the network to focus on the most relevant and salient features by encouraging it to learn more compact and abstract representations deeper into the network.

### 3.2 CONSTRUCTION OF CAPSULES

We model capsules as Monte Carlo sampled log-normally distributed continuous random variables, as this allows us to define the agreement between lower-level capsules (hereafter referred to as child capsules) and higher-level capsules (hereafter referred to as parent capsules) using information theory. We can assume such log-normal distributions because, due to the central limit theorem, the internal activation of artificial neurons can often be regarded as an approximate Gaussian distribu-

tion, and when feeding this into an exponential activation function $e^x$, this results in a log-normal distribution.

However, in our implementation we use the softplus activation function $\zeta(x)$ (Equation 1) instead of the exponential function in the capsule layers since it was empirically observed to perform better and to be more stable. We assume that this is because of numerical instabilities due to very large output values of the exponential function. Note that softplus applied to a Gaussian distribution does not only have a similar appearance of a log-normal distribution in the same co-domain $(0, +\infty)$, but that it is a good approximation of the exponential function for $x$ being small or negative, as the quotient of $\frac{\zeta(x)}{e^x}$ quickly converges to 1 for $x \to -\infty$. Moreover, the softplus activation function is differentiable and well-behaved.

$$\zeta(x) = log(1 + exp(x)) \tag{1}$$

An additional motivation behind treating capsules as Monte Carlo sampled continuous random variables is to generate multiple samples from the input space to approximate the true posterior distribution of object instantiation parameters.

Child capsules, $\Psi$, are constructed using 2D depthwise convolutions. Therefore, each child capsule $\psi \in \Psi$ is derived from just one input feature map; this formulation explicitly enforces each child capsule to be a Monte Carlo sampled continuous random variable that represents the probability distribution over the pose and features of a specific part or object its respective input feature map encodes. By letting each child capsule to attend to some subregion of only one input feature map, the child capsules can capture the variability and uncertainty of object descriptors more accurately and, therefore, the parent capsules are able to filter out noisy child capsules more easily, at least in theory. A further reasoning for using 2D depthwise convolution to construct the child capsules is the following: Since capsules are basically 3D tensors that are created from 2D tensors (regular 2D convolution), by applying a different filter to each input feature map, the resulting capsules do not break the convention of channels in convolutions; i.e. the 3D tensor has $N$ sets of $K \times K$ capsules of length $D$.

The kernel size for the 2D depthwise convolution is $7 \times 7$ with a stride of 2, and the channel multiplier is the length of the child capsules, which we set to 8, as per Sabour et al. (2017). Thus, the output of the 2D depthwise convolution is a $(K, K, D \times N)$ tensor, where $K$ is the feature map height and width respectively, $D$ is the length of the child capsules, i.e. the depth multiplier, and $N$ is the number of input channels. The output tensor is then reshaped to $(D, 1, 1, K \times K \times N)$ to enable the construction of parent capsules. Therefore, 2D depthwise convolution conveniently allows to move from an N-dimensional tensor to an N+1-dimensional tensor.

Parent capsules, $\Omega$, are derived from child capsules through a 3D transpose convolution. Whereas each child capsule attends to a subregion of one input feature map, each parent capsule attends to all child capsules, thereby aggregating the information the child capsules encode. The motivation behind using a 3D transpose convolution is that it enables to go from a $(D, 1, 1, N)$ tensor to a $(D', 1, 1, N')$ tensor, where $D' >> D$ and $N' << N$. $D'$ is 16, as defined by Sabour et al. (2017), and $N'$ is the number of classes. The kernel size for the 3D transpose convolution is $5 \times 1 \times 1$ and the strides are $(2, 1, 1)$ in order to double the depth dimension, i.e. capsule size.

### 3.3 ENTROPY VOTING

As each $\omega \in \Omega$ is conditioned on all $\psi \in \Psi$, we know that there is a latent joint distribution $p(\Psi, \Omega)$, which allows to compute the agreement between child and parent capsules using information theory. The parent capsule $\omega \in \Omega$ with the highest average entropy score w.r.t. to all $\psi \in \Psi$ is considered the correct prediction.

We define entropy voting as

$$f(\Psi, \Omega) = h(\Psi) + h(\Omega) - D_{KL}(\Psi||\Omega), \tag{2}$$

with $h(\Psi)$ and $h(\Omega)$ being the marginal differential entropy terms and $D_{KL}(\Psi||\Omega)$ being the Kullback-Leibler (KL) divergence of $\Psi$ from $\Omega$. The log-normal marginal differential entropy term for random variable $x$ is given by

$$h(x) = \mu - \frac{1}{2}\ln(2\pi e \sigma^2). \tag{3}$$

Thus, to maximize $f(\Psi, \Omega)$, we want to maximize the log-normal marginal differential entropy terms (Equation 3) $h(\Psi)$ and $h(\Omega)$, while minimizing $D_{KL}(\Psi||\Omega)$; the higher the marginal entropy terms $h(\Psi)$ and $h(\Omega)$ are, the more information the underlying log-normal distributions carry, and the lower the $D_{KL}(\Psi||\Omega)$ term is, the better some $\omega \in \Omega$ describes all $\psi \in \Psi$ on average. Since capsules are bound to encode shared parameters about pose, texture, color, etc., as they are global properties, by maximizing $h(\Psi)$ and $h(\Omega)$, the capsules are encouraged to capture object-specific properties better, because maximizing differential entropy is basically the same as maximizing variance, which makes the underlying distribution spread out.

$$D_{KL}(P||Q) = log\frac{\sigma_1}{\sigma_0} + \frac{\sigma_0^2 + (\mu_0 - \mu_1)^2}{2\sigma_1^2} - \frac{1}{2} \tag{4}$$

The Kullback-Leibler divergence (also known as relative entropy) $D_{KL}(P||Q)$ - as defined for two univariate distributions - is used to model agreement between the actual distribution $P$ and a reference distribution $Q$. Thus, relative entropy can be interpreted to measure how much information is lost when approximating $P$ with $Q$; a relative entropy of 0 can be considered to indicate that the two distributions $P$ and $Q$ have identical quantities of information and thereby no information is lost when approximating $P$ using $Q$. Therefore, KL divergence sufficiently models agreement between child capsules $\Psi$ (the actual distribution) and parent capsules $\Omega$ (the reference distribution) in our formulation.

$$\sigma(x) = \frac{1}{1 + e^{-x}} \tag{5}$$

Finally, to obtain the prediction probabilities, the output of $f(\Psi, \Omega)$ is pushed through the sigmoid function (Equation 5). The sigmoid function is chosen for two reasons: 1) the co-domain of $f(\Psi, \Omega)$ maps well to the domain of the sigmoid function, and 2) sigmoid is non-mutually exclusive, meaning that it allows for multi-class classification.

---

**Algorithm 1** Entropy voting.

---

1: **procedure** ENTROPY VOTING($X, Y$)
2:      $\forall \psi \in \Psi : \alpha_i \leftarrow h(\psi)|_{i=1}^{|\psi|}$        ▷ Computes Eq. 3
3:      $\forall \omega \in \Omega : \beta_j \leftarrow h(\omega)|_{j=1}^{|\omega|}$        ▷ Computes Eq. 3
4:      $\forall \psi \in \Psi; \forall \omega \in \Omega : \delta_{ij} \leftarrow D_{KL}(\psi||\omega)$        ▷ Computes Eq. 4
5:      $z_j \leftarrow \frac{1}{|\Psi|}\sum_i \sum_j \alpha_i + \beta_j - \delta_{ij}$
6:      $\hat{y}_j \leftarrow \sigma(z_j)$        ▷ Computes Eq. 5
7:      **return** $\hat{y}_j$
8: **end procedure**

---

Note that the entropy voting function $f(\Psi, \Omega)$ is strongly motivated by mutual information $I(\Psi; \Omega)$: Since $D_{KL}(\Psi||\Omega) = h(\Psi, \Omega) - h(\Psi)$ and $I(\Psi; \Omega) = h(\Psi) + h(\Omega) - h(\Psi, \Omega)$, one can rewrite Equation 2 as $f(\Psi, \Omega) = h(\Psi) + I(\Psi; \Omega)$. Thus, maximizing $f(\Psi, \Omega)$ means both maximizing the marginal differential entropy of $\Psi$ and the mutual information of $\Psi$ and $\Omega$. While the former guarantees that the underlying distribution carries as much information as possible, the latter ensures agreement between $\Psi$ and $\Omega$.

One big advantage of using entropy voting besides its theoretical justification is that it can be computed very efficiently due to its lack of mixed entropy terms; computing the relative entropy of two univariate distributions is computationally tractable as it can be done using their respective moments (Belov and Armstrong, 2011).

## 4 EXPERIMENTS

### 4.1 EXPERIMENTAL SETUP

We conduct our experiments on the MNIST (LeCun et al., 2010), CIFAR10 (Krizhevsky, 2009), SVHN (Netzer et al., 2011) and smallNORB (LeCun et al., 2004) datasets as these are the standard datasets in research on capsule networks. We use the same training setup for all datasets; a batch size of 128, Adam optimizer with a learning rate of 0.001, and a learning rate scheduler that halves the learning rate on plateau after 5 epochs. We train both with augmentations and completely without augmentations. We augment the images with random affine transformations; scale, translation, rotation, and shear, all within a $\pm 20\%$ range. Affine transformations are basic data augmentation techniques that apply geometric operations to the image without changing its content, e.g. as opposed to cropping. As for the hardware, we use an NVIDIA V100 GPU for training and testing.

The MNIST dataset contains a total of 70,000 grayscale images of handwritten digits. The images are of size 28x28 pixels. Each image represents a single digit ranging from 0 to 9, with the digit being in the center of the image. The training set consists of 60,000 images, while the test set contains 10,000 images.

CIFAR10 consists of a collection of labeled images representing various objects and scenes. It contains 60,000 color images in RGB format. Each image has a size of 32x32 pixels and is divided into 10 different classes. The training set contains 50,000 images, while the test set contains 10,000 images.

The SVHN dataset is an image digit recognition dataset of over 600,000 RGB images of size 32x32. The labels range from 0 to 9, representing the digits from zero to nine. Each image contains one or more digits. For training, we use 73,257 images and for testing 26,032 images.

The smallNORB dataset consists of grayscale images of 3D objects. The objects belong to five different categories, and each object was imaged by two cameras under 6 lighting conditions, 9 elevations, and 18 azimuths. The dataset is split evenly between training and testing sets, with 24,300 images each. The training set contains 5 instances (4, 6, 7, 8 and 9) of each category, and the test set the remaining 5 instances (0, 1, 2, 3, and 5).

For comparison we reimplemented CapsNet by Sabour et al. (2017) according to the description given in the paper, used the official implementation of Efficient-CapsNet by Mazzia et al. (2021), and reimplemented HVC-CapsNet (for the lack of better nomenclature) by Byerly et al. (2021), also as per the description given in the paper. A direct comparison with the results given in their papers is problematic as different augmentation methods are used, which is one of the issues capsule networks were designed to address. Furthermore, since two of the three baselines - CapsNet and Efficient-CapsNet - benefit from a reconstruction network as a regularizer, we tested our model with and without it. That being said, the parameter counts in Table 1 exclude the reconstruction regularizer for simplicity of comparison.

The reconstruction regularizer is a simple three-layer feedforward network with 512 and 1024 units in the first two layers, respectively, and both layers use the ReLU activation function. The output layer uses the sigmoid activation function, and the number of units is equal to the number of pixels in the input image. In the loss function, the sum of squared penalties between the normalized input pixel values and the reconstruction output is minimized. The penalty term is scaled down by a factor of 0.0005 and added to the overall loss (Sabour et al., 2017).

### 4.2 RESULTS

As can be seen in Table 1 below, our approach outperforms all of the three baselines on every dataset, not only in the error rates but also in inference speed, given in frames per second (FPS). Compared to CapsNet the number of parameters is one order of magnitude lower although it does not yet reach the low parameter count of Efficient-CapsNet. Regardless, our method still achieves a higher FPS. This is mainly due to differences in the voting procedure, as attention-based methods tend to be computationally more expensive.

For the smallNORB experiments we use two different input sizes for better comparison to the baselines. In the case of CapsNet, smallNORB images are first resized to $48 \times 48$ and then patches of

Table 1: Results from our implementations. Test error % averages and standard deviations of top-5 trials. No ensembles used. [†] input size $48 \times 48$. [‡] input size $32 \times 32$.

| DATASET | METHOD | PARAMS | FPS | W/O AUG | W/ AUG |
|---------|--------|--------|-----|---------|--------|
| MNIST | Our method w/o recon. | 643 | **161** | **0.28**$_{\pm 0.008}$ | **0.22**$_{\pm 0.007}$ |
| | Our method w/ recon. | 643 | 120 | 0.34$_{\pm 0.03}$ | 0.31$_{\pm 0.03}$ |
| | Sabour et al. (2017) | 6806 | 86 | 0.57$_{\pm 0.02}$ | 0.35$_{\pm 0.03}$ |
| | Mazzia et al. (2021) | 161 | 112 | 0.41$_{\pm 0.02}$ | 0.37$_{\pm 0.06}$ |
| | Byerly et al. (2021) | 1237 | 66 | 0.48$_{\pm 0.02}$ | 0.38$_{\pm 0.02}$ |
| CIFAR10 | Our method w/o recon. | 680 | **157** | **16.32**$_{\pm 0.1}$ | **13.83**$_{\pm 0.08}$ |
| | Our method w/ recon. | 680 | 117 | 17.16$_{\pm 0.34}$ | 15.72$_{\pm 0.43}$ |
| | Sabour et al. (2017) | 7993 | 83 | 34.47$_{\pm 0.17}$ | 30.03$_{\pm 0.69}$ |
| | Byerly et al. (2021) | 1453 | 64 | 22.35$_{\pm 0.11}$ | 21.89$_{\pm 0.62}$ |
| SVHN | Our method w/o recon. | 680 | **157** | **6.26**$_{\pm 0.03}$ | **4.79**$_{\pm 0.04}$ |
| | Our method w/ recon. | 680 | 117 | 6.70$_{\pm 0.19}$ | 5.22$_{\pm 0.41}$ |
| | Sabour et al. (2017) | 7993 | 83 | 8.75$_{\pm 0.07}$ | 6.9$_{\pm 0.43}$ |
| smallNORB 48x48 | Our method w/o recon. | 774 | **152** | **6.25**$_{\pm 0.53}$ | **5.92**$_{\pm 0.46}$ |
| | Our method w/ recon. | 774 | 113 | 7.13$_{\pm 0.47}$ | 7.18$_{\pm 0.81}$ |
| | Mazzia et al. (2021)[†] | 151 | 108 | 10.58$_{\pm 1.15}$ | 9.25$_{\pm 0.27}$ |
| smallNORB 32x32 | Our method w/o recon. | 646 | **158** | **7.57**$_{\pm 0.42}$ | **7.16**$_{\pm 0.39}$ |
| | Our method w/ recon. | 646 | 118 | 8.55$_{\pm 0.58}$ | 9.54$_{\pm 0.23}$ |
| | Sabour et al. (2017)[‡] | 6640 | 84 | 12.88$_{\pm 0.31}$ | 11.6$_{\pm 0.28}$ |

$32 \times 32$ are cropped out (Sabour et al., 2017), whereas in the case of Efficient-CapsNet, the images are resized to $64 \times 64$, after which patches of $48 \times 48$ are cropped out (Mazzia et al., 2021). We noticed that CapsNet performs suboptimally on inputs sizes larger than $32 \times 32$, and that Efficient-CapsNet is limited to only two different input sizes, i.e. $28 \times 28$ and $48 \times 48$ (hence experiments only on MNIST and smallNORB). We tested our own method on both input sizes for smallNORB, and outperformed both baselines.

We also tested our model on the original smallNORB with input image size of $96 \times 96$. However, from all the datasets, training on the original smallNORB resulted in the most unstable test performance, especially with data augmentations, which we suspect might be due to insufficient capsule sizes; the higher the resolution of the input, the bigger the capsules ought to be to better capture the more granular pixel information. We tested this hypothesis empirically, and observed an improvement in stability (i.e. a decrease in standard deviation of test errors) of predictive performance on the original smallNORB (without augmentations) when doubling and quadrupling the sizes of both child and parent capsules. That being said, however, if to look at top-5 test results only, keeping the capsule sizes as-is - and not using any data augmentations - resulted in the lowest average test error % of $6.57_{\pm 1.06}$ (as opposed to $8.97_{\pm 0.71}$ with data augmentation), with the lowest test error being 4.72%.

Note that the test results given in the baseline papers (see Table 2) differ from our own experiments, mainly due to their use of different data augmentations and experimental setup. Moreover, since one of the main advantages of capsule networks from a theoretical point of view is the ability to learn robust abstract representations without the need for data augmentation, we believe that it is best to compare different capsule network approaches without any data augmentation at all, but provide results with data augmentation as well. However, while Sabour et al. (2017), Mazzia et al. (2021),

Table 2: Test error % reported in other CapsNet works. Best only, if no standard deviation is given. All results use some kind of data augmentation or reconstruction regularizer. Results obtained with ensembles in italics.

| METHOD | MNIST | CIFAR10 | SVHN | SmallNORB |
|---|---|---|---|---|
| Sabour et al. (2017) | $0.25_{\pm 0.005}$ | *10.6* | 4.3 | 2.7 |
| Hinton et al. (2018) | - | - | - | 1.4 |
| Zhang et al. (2018) | - | 5.19 | 1.79 | - |
| Hahn et al. (2019) | - | $7.86_{\pm 0.12}$ | $3.12_{\pm 0.13}$ | $15.91_{\pm 1.09}$ |
| Rajasegaran et al. (2019) | 0.28 | *7.26* | *2.44* | - |
| Ribeiro et al. (2020) | - | $11.2_{\pm 0.09}$ | $3.9_{\pm 0.06}$ | $1.6_{\pm 0.06}$ |
| Zhang et al. (2020) | 0.38 | 14.3 | - | 2.2 |
| De Sousa Ribeiro et al. (2020) | $0.28_{\pm 0.01}$ | - | $7.0_{\pm 0.15}$ | $1.4_{\pm 0.09}$ |
| Mazzia et al. (2021) | $0.26_{\pm 0.0002}$ | - | - | $2.54_{\pm 0.003}$ |
| Byerly et al. (2021) | *0.13* (0.17) | $11.08_{\pm 0.002}$ | - | - |

and Byerly et al. (2021) use different augmentations compared to the augmentations we use (they also differ from each other), we use the same augmentation setup for all methods in our experiments.

It is also worth to note that the biggest discrepancy in terms of the results from our experiments and what was reported by the authors in their respective papers is between our implementation of HVC-CapsNet (Byerly et al., 2021) as tested on MNIST, and the results reported by Byerly et al. (2021) on MNIST. This might be due to the specific data augmentation techniques used by the authors, which include cropping a $4 \times 4$ patch of pixels from the input image, for example. However, it can also be that there is, for example, a disconnect between the description of the model as provided in the paper and the implementation details of the model used in their experiments. Regardless, as can be seen in Table 1, of the chosen baselines, HVC-CapsNet performs the poorest on MNIST using generic data augmentations. That being said, none of the baseline implementations achieved performance similar to what was reported in their respective papers.

For completeness, Table 2 also gives the results of the three baseline methods as well as additional ones as given in the respective papers. Note, that the exact results cannot directly be compared to each other since they use different augmentation techniques, some use a reconstruction regularizer, and some results are based on ensembles. Moreover, most of the results are best only and not averages with some given standard deviation. Nevertheless, our proposed method, coupled with simple affine transformations as data augmentation, achieves better performance on MNIST than all approaches but Byerly et al. (2021), which we were not able to reproduce using our implementation of their approach.

However, when considering other datasets, our method does not exhibit comparable performance to the leading approaches reported in the table, which may be because we used the same model for all datasets, whereas others tweaked their model to the dataset in question - or used ensembles - coupled with extensive data augmentation techniques. We chose to not tweak our model to each dataset or use an ensemble because we were interested in exploring the generalization ability of our approach in order to investigate the potential of capsules as-is. That is also why we deem it important to measure the performance of capsule networks without data augmentation.

This brings forth an important question regarding the effectiveness of data augmentation in the context of capsule networks. Considering the fundamental objective of capsules, which is to acquire robust and equivariant object representations and instantiation parameters, the use of data augmentation becomes a matter of inquiry. Capsule networks aim to address the limitations associated with the need for extensive data augmentation, which can be computationally strenuous. One of the primary challenges that capsules seek to overcome is the reliance on data augmentation techniques, and an exhaustive amount of data in general, which are computationally expensive.

## 5 LIMITATIONS

As can be seen in Table 1, our method performs best on MNIST, arguably the simplest of the four datasets used in the experiments. Intuitively this makes sense, as the digits from 0 to 9 are sufficiently distinct to be trivially recognizable, at least when written clearly, and therefore the model should be able to discover log-normal distributions unique to each digit. Rather unsurprisingly, CIFAR10 proved to be the most difficult dataset to do well on. While CIFAR10 can probably be considered the most complex of the datasets used in the experiments, it does raise a question about the relationship between image resolution and its content; the more complex the content of an image is, the more useful a higher resolution is.

Thus, from the experimental results one might conclude that our method performs well only on simpler datasets, such as MNIST. However, given the resolution of the datasets, it is difficult to draw any concrete conclusions in that regard. Furthermore, since our method performs better on smallNORB with the images resized to a higher resolution rather than to a lower one, it does indicate that our method might benefit from a higher resolution in general. This makes sense, as each pixel is basically one unit of information, so with a higher resolution, our method is able to sample from a more granular input space and therefore better approximate the true posterior distribution of object instantiation parameters.

Although our method does not perform as well on the original smallNORB as it does on the $48 \times 48$ downscaled variant of it, the average test error % without data augmentation on the original small-NORB indicates that capsules might indeed be able to generalize better on higher resolution images, and in turn suffer from data augmentation in such situations. However, our method was rather unstable (a high standard deviation of test errors) in this particular case, and we observed the standard deviation decrease and stabilize when increasing capsule sizes. Nonetheless, bigger capsules did not offer better predictive performance in the end, so there is a trade-off between stability and predictive accuracy when it comes to the original smallNORB dataset, and potentially other higher resolution datasets, too.

## 6 CONCLUSION

In this work, we proposed a non-iterative voting procedure between child and parent capsules that is motivated by information theory. We proved empirically that through discriminative learning, our method is able to learn representations that generalize sufficiently well, even without data augmentation, by maximizing the marginal differential entropies of all capsules while minimizing the relative entropy between child and parent capsules.

Thus, our work provides two contributions to the field of capsule networks; (1) an interpretation of capsules and the voting procedure that is firmly rooted in information theory, which also allows for the use of a differentiable activation function for capsules, and (2) the use of 3D transpose convolution to derive parent capsules from child capsules so that changes in dimensions and sizes between the capsules are taken into account, instead of using workaround matrix operations to achieve the same effect.

Although our proposed method exhibits degradative performance on more complex datasets, the experiments also indicate that it benefits from a larger input size and, to some extent, larger capsule size. Hence, there is room for research on capsule networks with respect to more complex, higher resolution images, especially without data augmentation. In general, methods that aim to make data augmentation unnecessary should receive more attention in the research community as data augmentation is a roundabout way of addressing one of the most fundamental challenges in computer vision.

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
