# OpenReview forum: "Entropy Voting Between Capsules"
_ICLR.cc/2024/Conference — Submitted to ICLR 2024_

### Official Review · Reviewer_VTJB · 2023-10-28

**Soundness:** 2 fair
**Presentation:** 2 fair
**Contribution:** 1 poor
**Rating:** 3
**Confidence:** 4

**Summary:**

The authors view capsule neural network (henceforth ConvNets) routing through the lenses of information theory. They propose a non-iterative routing mechanism based on entropy that deals with the voting process. Despite having some nice properties (it is non-iterative for one) it does not seem to actually work well in practice, at least not with respect to other methods that have been proposed since dynamic routing was proposed back in 2017. \
I do not think that SOTA is or should be the main criterion but the lack of performance overall might point to an inefficient routing mechanism with respect to how lower and higher capsules are learned and routed through.

**Strengths:**

This is a simple non-iterative routing process that is straightforward to implement and is relatively efficient with respect to number of parameters. Had this paper appeared 4 years ago it would have been a very good CapsNet formulation but nowadays CapsNets with 60K-300K outperform this paper on pretty much every dataset.

**Weaknesses:**

The main limitation is that in absence of SOTA results the paper lacks a more technical or theoretical interpretation of how routing based on entropy voting unravels interesting CapsNets properties that might point to an interesting future direction. By that I mean that if the authors present a more in depth analysis, perhaps through visualisations and capsule activations, of what is taking place when routing with entropy, it might provide some insights as to how future Capsnet models could be improved to the point of surpassing current models.

**Questions:**

1) Could you please provide a more in depth analysis as to what advantages entropy might offer to capsules beyond the technical bit? For instance, other papers have used attention to capture local features and local-global context, or leveraged pruning to reduce parameters etc. What does entropy offer to how the part-whole relationships are learned?
2) Any insights into why performance lags with respect to pretty much most of the more recent papers? Is there anything fundamental that hinders that and/or does this imply that entropy might not be the way forward?

---

### Official Review · Reviewer_Y23x · 2023-11-01

**Soundness:** 2 fair
**Presentation:** 2 fair
**Contribution:** 2 fair
**Rating:** 3
**Confidence:** 2

**Summary:**

The paper proposes a new method for capsule networks called entropy voting, which is motivated by concepts from information theory. Key contributions are an information-theoretic interpretation of capsules and voting, and the use of 3D transpose convolutions for constructing capsules.

**Strengths:**

The voting procedure proposed avoids computationally heavy iterative routing, which is very practical.

**Weaknesses:**

I think there is insufficient experimentation to claim significant contributions, as author claims:

However, when considering other datasets, our method does not exhibit comparable performance to the leading approaches reported in the table, which may be because we used the same model for all datasets, whereas others tweaked their model to the dataset in question - or used ensembles - coupled with extensive data augmentation techniques. We chose to not tweak our model to each dataset or use an ensemble because we were interested in exploring the generalization ability of our approach in order to investigate the potential of capsules as-is. That is also why we deem it important to measure the performance of capsule networks without data augmentation.

This unfortunately produces incomplete work. Multiple models should be tested to validate your hypothesis. Cutting it short here is not persuasive.

**Questions:**

Did you complete any ablation study?

---

### Official Review · Reviewer_66co · 2023-11-01

**Soundness:** 2 fair
**Presentation:** 1 poor
**Contribution:** 2 fair
**Rating:** 3
**Confidence:** 5

**Summary:**

In this work, authors proposed a non-iterative voting procedure between child and parent capsules that is motivated by information theory. Through discriminative learning, this method is able to learn representations that generalize sufficiently well, even without data augmentation, by maximizing the marginal differential entropies of all capsules while minimizing the relative entropy between child and parent capsules. Authors also proposed to construct child capsules using depthwise convolutions and parent capsules using 3D transpose convolutions.

**Strengths:**

1. The capsule and voting process is explained through information theory, making the motivation for the proposed method clearer.
2. Through an entropy voting mechanism that differs from the traditional dynamic routing algorithm in capsule networks, iterations are avoided and the number of participants is greatly reduced.

**Weaknesses:**

1.	Section 3.1 should briefly introduce the proposed method in the network structure.
2.	It is not well explained why the lower-level capsules use depthwise convolution and the higher-level capsules use 3D transpose convolution.
3.	The method proposed in this paper is very different from the traditional capsule network in the way of generating high level capsules, which is suggested to be described in detail.
4.	There should be more details on how to connect the adjacent capsule layers and how and where to apply the proposed entropy voting method.
5.	Subsection 4.1 is too lengthy in introducing the dataset used for the experiments.
6.	The method of comparison in Table 1 is too old and there is only one comparison method on the SVHN dataset.
7.	No ablation experiments were performed and too few experimental results led to many conclusions being empirical and unconvincing. Suggest adding additional experimental results.

**Questions:**

1.	Section 3.1 should briefly introduce the proposed method in the network structure.
2.	It is not well explained why the lower-level capsules use depthwise convolution and the higher-level capsules use 3D transpose convolution.
3.	The method proposed in this paper is very different from the traditional capsule network in the way of generating high level capsules, which is suggested to be described in detail.
4.	There should be more details on how to connect the adjacent capsule layers and how and where to apply the proposed entropy voting method.
5.	Subsection 4.1 is too lengthy in introducing the dataset used for the experiments.
6.	The method of comparison in Table 1 is too old and there is only one comparison method on the SVHN dataset.
7.	No ablation experiments were performed and too few experimental results led to many conclusions being empirical and unconvincing. Suggest adding additional experimental results.

---

### Meta-Review · Area_Chair_zWoR · 2023-12-06

**Metareview:**

This paper offers a new perspective on capsule neural networks through an information-theoretic lens. It introduces entropy voting, a non-iterative routing mechanism based on entropy. The method is efficient and also generalizes well when the training data is limited. However, all reviewers expressed concerns that the experiments are insufficient to claim significant contributions. No state-of-the-art methods are compared in the paper, and there also lacks in-depth analyses of the model and experiments for ablation study. The authors didn’t respond during the rebuttal.

**Justification For Why Not Higher Score:**

The paper did not receive a higher score primarily due to unanimous concerns from reviewers about the insufficient experiments. The lack of comparisons with state-of-the-art methods and the absence of in-depth analyses and ablation studies further limited its evaluation. Additionally, the authors' lack of response during the rebuttal phase left these critical concerns unaddressed.

**Justification For Why Not Lower Score:**

N/A

---

### Decision · Program_Chairs · 2024-01-16

Reject